# Organization and Characterization of the Promoter Elements of the rRNA Operons in the Slow-Growing Pathogen *Mycobacterium kumamotonense*

**DOI:** 10.3390/genes14051023

**Published:** 2023-04-30

**Authors:** Ricardo Sánchez-Estrada, Oscar Méndez-Guerrero, Lázaro García-Morales, Jorge Alberto González-y-Merchand, Jorge Francisco Cerna-Cortes, María Carmen Menendez, María Jesús García, Lizbel Esperanza León-Solís, Sandra Rivera-Gutiérrez

**Affiliations:** 1Departamento de Microbiología, Escuela Nacional de Ciencias Biológicas, Instituto Politécnico Nacional (IPN), Prolongación de Carpio y Plan de Ayala S/N, Colonia Santo Tomas, Delegación Miguel Hidalgo, Ciudad de México 11340, Mexico; rsancheze1501@alumno.ipn.mx (R.S.-E.); lazaro.garcia@cinvestav.mx (L.G.-M.);; 2Departamento de Química Inorgánica, Escuela Nacional de Ciencias Biológicas, Instituto Politécnico Nacional (IPN), Prolongación de Carpio y Plan de Ayala S/N, Colonia Santo Tomas, Delegación Miguel Hidalgo, Ciudad de México 11340, Mexico; 3Departamento de Biomedicina Molecular, Centro de Investigación y de Estudios Avanzados del Instituto Politécnico Nacional, Ciudad de México 07360, Mexico; 4Departamento de Medicina Preventiva, Facultad de Medicina, Universidad Autónoma de Madrid, C. Arzobispo Morcillo, 4, 28029 Madrid, Spain

**Keywords:** *Mycobacterium kumamotonense*, rRNA operons, *rrn* promoters

## Abstract

The slow-growing, nontuberculous mycobacterium *Mycobacterium kumamotonense* possesses two rRNA operons, *rrnA* and *rrnB*, located downstream from the *murA* and *tyrS* genes, respectively. Here, we report the sequence and organization of the promoter regions of these two *rrn* operons. In the *rrnA* operon, transcription can be initiated from the two promoters, named P1 *rrnA* and PCL1, while in *rrnB*, transcription can only start from one, called P1 *rrnB*. Both *rrn* operons show a similar organization to the one described in *Mycobacterium celatum* and *Mycobacterium smegmatis*. Furthermore, by qRT-PCR analyses of the products generated from each promoter, we report that stress conditions such as starvation, hypoxia, and cellular infection affect the contribution of each operon to the synthesis of pre-rRNA. It was found that the products from the PCL1 promoter of *rrnA* play a pivotal role in rRNA synthesis during all stress conditions. Interestingly, the main participation of the products of transcription from the P1 promoter of *rrnB* was found during hypoxic conditions at the NRP1 phase. These results provide novel insights into pre-rRNA synthesis in mycobacteria, as well as the potential ability of *M. kumamotonense* to produce latent infections.

## 1. Introduction

*Mycobacterium kumamotonense* (*M. kumamotonense*) is a slow-growing nontuberculous mycobacterium (NTM) isolated for the first time from the sputum of a 57-year-old Japanese woman [1]. It has been reported that commercial molecular probes (AccuProbe System bioMérieux^®^, Marcy-l’Étoile, France and INNo-LiPA Mycobacteria v2 Innogenetics^®^, Ghent, Belgium), which align in the region of the genome that encodes the rRNA of the *Mycobacterium tuberculosis* complex (MTC), showed cross-reactivity with *M. kumamotonense* [2]. The misidentification of *M. kumamotonense* as a member of the MTC has serious clinical implications given that a patient infected with this species could be diagnosed with tuberculosis and treated with specific drugs for a long time and would be susceptible to several adverse secondary effects. Furthermore, *M*. *kumamotonense* shows resistance to antifimics such as isoniazid, pyrazinamide, and kanamycin [2].

Not only have infections caused by this NTM been reported to occur in both immunocompromised individuals (who are at higher risk of infection) and apparently healthy, immunocompetent individuals [3,4,5,6], but *M. kumamotonense* has also recently been isolated from a hospital environment [7], thus giving relevance to hospital-acquired infection (HAI).

Mycobacteria have been classified into two categories based on their growth rate (fast-growing and slow-growing). However, all of them are very closely related by the high levels of similarity (94% or more) between their 16S rRNA sequences [8]. To date, the molecular mechanisms used by mycobacteria to relate ribosome synthesis and the growth rate are not known in detail. Previously, it has been described that all mycobacteria have one *rrn* operon (*rrnA*) located downstream from the *murA* gene [9,10]; in addition, other mycobacterial species have a second operon (*rrnB*) located downstream of the *tyrS* gene [9,11]. A few mycobacterial species of the *Mycobacterium terrae* group (MTG), in which *M. kumamotonense* was also classified, carry two *rrn* operons per genome [12]. All the same, the sequence and organization of its two previously discovered *rrn* operons is not yet known.

In the present study, we report the organization of the promoter elements in the *rrnA* and *rrnB* operons of *M. kumamotonensis,* including partial sequences of the genes located upstream from each operon (*murA* and *tyrS,* respectively) and the DNA sequences of the intergenic regions present in both operons, highlighting the transcription starting points (*tsp*) located in the hypervariable multiple promoter regions (HMPR) based on the comparison of three mycobacterial species: (1) the slow-growing NTM *Mycobacterium celatum* (*M. celatum*), which is related to MTG; (2) the most extensively studied, fast-growing NTM *Mycobacterium smegmatis* (*M. smegmatis*); and (3) the major mycobacterial pathogen *Mycobacterium tuberculosis* (*M. tuberculosis*), as a model for all members of the MTC. Furthermore, it is shown how different stress conditions (lack of nutrients, hypoxia, and cellular infection) to which this bacterium can be subjected affect the expression of 16S rRNA, through preferential use of its *rrn* operons as well as promoter predilection.

## 2. Materials and Methods

### 2.1. Bacterial Strain and Growth Conditions

*M. kumamotonense* CST 7247, acquired from the DSMZ – German Collection of Microorganisms (Leibniz Institute, Munich, Germany), was grown at 37 °C in Dubos broth containing 0.1% (*w*/*v*) Tween 80 and 10% ADC [0.5% (*w*/*v*) bovine serum albumin, 0.2% (*w*/*v*) dextrose, and 0.004% (*w*/*v*) catalase] (Becton Dickinson, Cockeysville, MD, USA). Cells in the exponential phase were collected from cultures with an optical density (OD) at 600 nm of 0.4 units (day five). From the bacterial culture in the logarithmic phase, the non-replicative persistence 1 (NRP1) and non-replicative persistence 2 (NRP2) phases were established as previously described [13,14,15]. A parallel culture supplemented with methylene blue (1.5 μg/mL) was used as an indicator of oxygen depletion to establish the NRP phases [14,15]. The NRP1 phase was reached after six days of exposure to hypoxia (initial fading of methylene blue) and the NRP2 phase at day ten, under conditions of anaerobiosis (complete decolorization of methylene blue). Additionally, to validate the properly established hypoxic conditions, we evaluated the expression of the *groEL* (*Rv3417c*) and *ahpC* (*Rv2428*) genes that have been associated with the dormancy state of mycobacteria, using the primers and conditions shown in Appendix A.

In fact, cells in the stationary phase, also considered as a nutrient deprivation stage [16], were obtained when cultures reached an OD of 1.0 unit (day ten). In the same way, cell growth was evaluated via CFUs/mL measurements performed on Middlebrook 7H10 agar plates supplemented with 10% ADC (Becton Dickinson) using the microdrop technique [17] (Appendix A).

### 2.2. DNA Isolation

Total DNA was isolated from mycobacterial cells via the CTAB method [18]. Briefly, cells in the stationary phase were collected by centrifugation and resuspended in 1X TE buffer (10 mM Tris-HCl, 1 mM EDTA). This cell suspension was heated at 80 °C for 20 min. Then, 50 μL of 10 mg/mL lysozyme was added and the suspension was incubated for 1 h at 37 °C. Subsequently, 70 μL of 10% SDS and 5 μL of 20 mg/mL proteinase K were added, and the mixture was incubated at 65 °C for 10 min. Then, 100 μL of 5 M NaCl and 100 μL 0.3 M CTAB (both preheated at 65 °C) were added and the mixture was incubated at 65 °C for 10 min. Organic extraction was performed with 750 μL of phenol:chloroform:isoamyl alcohol (25:24:1) solution; after mixing and centrifugation, the aqueous phase was recovered, and DNA was precipitated with isopropanol at −20 °C overnight. Finally, the DNA pellet was resuspended in 1X TE buffer. DNA concentration and purity were analyzed with a NanoDrop (ND-1000; Thermo Scientific, Waltham, MA, USA). Integrity was determined by electrophoresis on 1.2% agarose gels.

### 2.3. Cloning and Sequencing of rRNA Operons of M. kumamotonense

For cloning experiments, genomic DNA of *M. kumamotonense* coding for *rrnA* and *rrnB* was amplified by PCR using RAC1 and RAC8 and TYRS3 and RAC8 mixtures, respectively (Appendix A). Amplified products were ligated into a pTZ57R/T vector (Thermo Scientific) through the T/A system and the recombinants plasmids were used to transform One Shot™ (Invitrogen, Waltham, MA, USA) *Escherichia coli* INVα′F′ competent cells according to the manufacturer’s instructions. Recombinants cells were selected on Luria agar plates with ampicillin (100 µg/mL) and X-gal (100 mg/mL). DNA of the recombinant clones was sequenced by Sanger dideoxy sequencing in an ABI PRISM™ 310 Genetic Analyzer (Applied Biosystems, Waltham, MA, USA) using the oligonucleotides mentioned above as sequencing primers. The sequences obtained were analyzed using BioEdit 7.0.5 software (Thomas Hall, USA).

### 2.4. Cell Line Culture and Infection

Human lung adenocarcinoma A549 cells (alveolar cells) (ATCC CCL-185) were cultured in DMEM (ATCC, Manassas, VA, USA) medium supplemented with heat-inactivated 10% fetal bovine serum (FBS, Gibco, New York, NY, USA) at 37 °C in a 5% CO_2_ atmosphere. Infection was performed in 24-well culture plates containing 1 × 10^6^ alveolar cells/well, that were infected with *M. kumamotonense* cells from the stationary phase (1 × 10^8^ bacteria/well), at a multiplicity of infection (MOI) of 100 (100:1) as previously reported in *M. tuberculosis* infections [19]. All plates were incubated for 2 h under standard conditions. Then, mycobacteria cells in the supernatant were collected to determine whether the microenvironment generated by the interaction with cells influenced the products generated from the rRNA promoters.

### 2.5. RNA Isolation and Synthesis of cDNA

Total RNA isolation from all the studied conditions (exponential, starvation stage, NRP1 and NRP2 phases, as well as alveolar cell infection) was carried out using the acid-phenol method [20]. Briefly, mycobacteria cells from each condition were collected by centrifugation and resuspended in lysis buffer (10% SDS, 1 M sodium acetate, 0.5 M EDTA) and mechanically disrupted using glass beads in a FastPrep-24 apparatus (MP Biomedicals, Santa Ana, CA, USA): 3 pulses at 6.5 m/s for 30 s (each pulse followed by a 5 min incubation on ice). Then, 500 μL of acid phenol preheated at 65 °C was added, the samples were mixed in a vortex and heated at 65 °C for 10 min, followed by incubation on ice for 5 min. After centrifugation, supernatants were recovered and two volumes of chloroform:isoamyl alcohol 24:1 were added, samples were mixed and centrifuged again, and the supernatants were recovered for RNA precipitation with two volumes of cold absolute ethanol and incubated at −20 °C overnight. Finally, RNA pellets were resuspended in nuclease-free water.

In order to remove any DNA contamination, RNA samples were treated with the TURBO DNA-free™ RNA purification kit (Thermo Scientific), and the absence of DNA was determined by qPCR, using *rrs*-Fw and *rrs*-Rv primers (Appendix A). The concentration and purity of RNA were evaluated using a NanoDrop, and its integrity was determined via denaturing gel electrophoresis in 2% agarose gels with added 0.1% chlorine [21].

cDNA synthesis was performed with 1.0 μg of RNA isolated from each condition, following the instruction of the SuperScript™ First-Strand Synthesis System for RT-PCR kit (Thermo Scientific).

### 2.6. Rapid Amplification of cDNA Ends (RACE)

According to the manufacturer’s instructions, 5′-RACE was carried out with the second-generation 5′/3′-RACE kit (Roche, Basel, Switzerland). Briefly, the synthesis of the first-strand cDNA was carried out using 2 μg of DNA-free RNA from *M. kumamotonense* with the RAC8 primer (Appendix A). In the first 5′-RACE-PCR, an aliquot of 5 μL of the first strand reaction (cDNA products) was amplified using the cR103 reverse primer (Appendix A) and the forward anchor primer, provided by the kit. Both primers were used at a concentration of 20 μM. The amplification program was as follows: 35 cycles of 15 s at 94 °C, 30 s at 55 °C, and 60 s at 72 °C. In the second 5′-RACE-PCR, an aliquot of 5 μL from the first 5′-RACE-PCR was amplified using the JY15 reverse primer (Appendix A) at 20 μM and the forward oligo(dT)-anchor primer provided by the kit. The second amplification program was performed under the same conditions as mentioned above, and three different PCR products were obtained (respectively correlated with a promoter) (Appendix A); each of them was purified with the QIAquick Gel Extraction kit (QIAGEN, Hilden, Germany) and sequenced by Sanger dideoxy sequencing.

### 2.7. Analysis of Precursor rRNA (pre-rrn) by RT-qPCR

The absolute quantification of the amounts of pre-*rrn* products from each promoter PCL1, P1 *rrnA*, or P1 *rrnB*, as related to their contribution to the expression of 16S RNA, was conducted with specific primers (Appendix A), using the 2XRotor-Gene SYBR Green PCR Master Mix (QIAGEN) and the Rotor-Gene Q instrument (QIAGEN). Quantification was performed five times using RNA isolated from three different culture batches of each condition. Standard curves were performed from 10-fold serial dilutions of known copies of *M. kumamotonense* CST 7247 genomic DNA (from 10^3^ to 10^7^ copies). In each condition, Ct values were interpolated to the standard curve for each pair of primers to obtain absolute quantification in copies/μg of RNA.

Expression of 16S rRNA was obtained by amplification with *rrs*-Fw and *rrs*-Rv primers (Appendix A, which amplify the region of 202 pb between nucleotides 973 and 1175 of the 3′ end of 16S rDNA), P1 *rrnA* expression was obtained by amplification with P1-Fw and P1-Rv primers (Appendix A), and P1 *rrnB* expression was obtained by amplification with P2-Fw and P2-Rv primers (Appendix A).

To obtain the specific number of copies generated from the PCL1 promoter, the total expression of 16S rRNA obtained as the number of copies/μg of RNA was considered as 100% of the expression, and then we subtracted the expression obtained from the P1 *rrnA* and P1 *rrnB* promoters, obtaining the expression of PCL1 as a result, as indicated below.
16S rRNA − (P1 *rrnA* + P1 *rrnB*) = PCL1

The absolute expression of the pre-*rrn* products generated from each PCL1 promoter, P1 *rrnA* or P1 *rrnB*, during the exponential and starvation stages, NRP1, NRP2, and alveolar cell infection was normalized using the total expression of 16S rRNA.

### 2.8. Statistical Analysis

For statistical analysis, two-way ANOVA followed by Tukey’s multiple comparison test was performed using GraphPad Prism 9.5.0 (GraphPad Software Inc., San Diego, CA, USA). Data represent the mean ± standard deviation (SD), and values of *p* < 0.05 and *p* < 0.005 were considered statistically significant.

### 2.9. Alignment of Sequences

DNA sequences of the *rrn*A operon of *M. kumamotonense* (KT878832.1, data obtained in this work), *M. celatum* (EF613279.1), *M. smegmatis* (X87943.1), and *M. tuberculosis* (X58890.1) were obtained from the GenBank Database [22] and aligned with the T-Coffee multiple sequence alignment program [23]. In such a way, DNA sequences of the *rrnB* operon of *M. kumamotonense* (OQ344267, data obtained in this work), *M. celatum* (EF613280.1), and *M. smegmatis* (U09862.1) were obtained and aligned. On the other hand, the nucleotide basic local alignment search tool (BLASTn) [24] was used to compare the sequences of the HMPRs of the *rrnA* and *rrnB* operons of *M. kumamotonense*.

## 3. Results

### 3.1. Organization of the rRNA Promoters

Previously, using the restriction fragment length polymorphism (RFLP) pattern of the 16S rDNA gene of *M. kumamotonense*, it was found that this bacterium carries two copies of the *rrn* operon per genome [12].

Here, analyzing the identities of the upstream genes, it was found that *M. kumamotonense* has one *rrnA* and one *rrnB* operon per genome similar to 12 other distantly related mycobacteria, including *M. celatum* and *M. smegmatis* [25]. Comparison of the HMPR and signature sequences of the 16S rRNA allowed us to elucidate the scheme of each promoter region. A schematic representation of the promoter region of each one of the two *rrn* operons of *M. kumamotonense* is summarized in Figure 1A and Figure 2A, respectively, as well as the transcription starting points (*tsp*) for the two operons identified by 5′-end RACE experiments and by comparison with other putative promoter sequences of NTM (Figure 1B and Figure 2B).

Transcription of the *rrnA* operon was found to be regulated by two promoters, designated P1 and PCL1 [9]. As shown in Figure 1A, the promoter P1 *rrnA* was found downstream from the end of the *murA* coding region near the stop codon. While the promoter PCL1 was located near the conserved leader sequence-1 (CL1 motif), which is a feature of *rrnA* operons [10]. On the other hand, the *rrnB* operon of *M. kumamotonense* was found to be only regulated by a single promoter, designated P1 *rrnB*. As shown in Figure 2A, promoter P1 *rrnB* was localized downstream from the *tyrS* gene, which is a feature of *rrnB* operons [9]. All mycobacterial species possessing two *rrn* operons per genome have one *rrn*A and one *rrn*B operon, regardless of the phylogenetic distances [25].

Analysis of the HMPR sequences showed a mean of 37% polymorphism between the *rrn* operons of *M. kumamotonese* and the *rrn* operons of the mycobacterial species analyzed (Figure 1B and Figure 2B). Comparing the HMPRs sequences of the *rrnA* and *rrnB* operons of *M. kumamotonense*, only 18% of coverage was observed between them.

Mycobacteria have developed strategies, including gene dosage and differential gene regulation, in order to maximize the efficiency and flexibility of rRNA synthesis with a minimal number of *rrn* operons. Control of the expression of the *rrnB* operon by a single promoter may reflect the fact that the *rrnB* operon is younger than *rrnA*, and P1 *rrnB* promoter duplication has not occurred yet [10]. Therefore, understanding how mycobacteria use these strategies in response to different conditions and stimuli should provide new insights into how this important genus of bacteria modify its rRNA synthesis according to distinct environmental conditions.

### 3.2. Expression of the Genes Related to Dormancy

An upregulation of the dormancy-related genes was found, studied from the low-oxygen stress conditions. Appendix A shows that *ahpC* was upregulated in both stages compared to the exponential condition, while *groEL* expression was only upregulated at the NRP1 phase supporting the establishment of the dormancy conditions of the *M. kumamotonense*.

### 3.3. Expression of the Pre-rrn Products under Stress Conditions

Expression of the pre-*rrn* products under stress growth conditions is summarized in Figure 3. It was found that all products of the *rrnA* promoters (PCL1 and P1 *rrnA*) presented the highest expression in all studied phases of growth. However, the PCL1 promoter had a more prominent role in 16S rRNA synthesis than the P1 promoter (Figure 3).

By contrast, the expression of the P1 *rrnB* promoter was the lowest in all studied conditions, mainly in the stationary and NRP2 phases. Interestingly, the highest quantification of P1 *rrnB* pre-*rrn* products was found during the NRP1 phase (Figure 3).

We also studied how the interaction within eukaryotic host cells affects rRNA synthesis [19]. We demonstrated that the microenvironment generated within the cells affects the promoters’ contributions to pre-rRNA synthesis. Compared with exponential conditions, the pre-*rrn* products coming from the P1 *rrnB* promoter increased, while PCL1 pre-*rrn* products decreased, both in a significant way (Figure 3).

## 4. Discussion

### 4.1. Arrangement of the rrnA and rrnB Promoters

Mycobacterial gene conversion is thought to be involved in *rrn* operon evolution [26]. In mycobacteria, the number of *rrn* operons per genome appears not to be correlated with the growth rate. However, mycobacterial species differ in the number of the rRNA operons. In *M. tuberculosis*, the *rrnA* operon is regulated by two tandem promoters, whereas in *M. chelonae* this regulation is controlled by five promoters [10]. Mycobacteria such as *M. celatum* or *M. smegmatis* have a second operon called *rrnB*, which is regulated by a single promoter [25,27].

Previously, Menéndez and co-workers (2014) demonstrated that *M. kumamotonense* has two *rrn* operons per genome. Moreover, they identified two putative sequences that could explain the detected cross-hybridization with *M. tuberculosis* [12]. Here, it was found that these two species have a high percentage of similarity when comparing their HMPR organization of the *rrn*A operons (93%), pointing out the cross identification between them as previously described [2].

We corroborated that *rrnA* operons of *M. kumamotonensis* and *M. tuberculosis* are regulated by two promoters (P1 and PCL1) (Figure 1B), making it difficult to correctly differentiate between these two species considering only this operon. A feature of the P1 *rrnA* promoter is the uncertainty regarding the location of its −35 box; nevertheless, the PCL1 promoter is characterized by the presence of the CL1 motif downstream from its −10 box [28], which promotes the formation of a transcription initiation complex [11]. The set of −10 and −35 boxes is typical of promoters requiring the initiation factor sigma 70 [10]. These promoter sequences do not overlap, indicating that each promoter might conform an initiation complex with *rpo* and sigma factors, independently [10] allowing the evaluation of its contribution to rRNA synthesis, as discussed below.

On the other hand, the *rrnB* operon was found to be regulated by a single promoter, P1 *rrnB* (Figure 2A). The HMPR organization of the *rrnB* operons in *M. kumamotonense* and *M. celatum* is 78% similar, compared to 83% similarity with *M. smegmatis*. The P1 *rrnB* promoter is characterized by a CL2 motif sequence (Figure 2B), coding for the putative BoxA element cataloged as an antitermination element of the leader region of the *rrn* operons, which is highly conserved between all mycobacterial operons, including *rrnA* [11,27]. However, since it does not have any motif that promotes the formation of a transcription initiation complex, as the CL1 motif of *rrnA* operons does, it could be the cause of this operon contributing less to rRNA synthesis. Unfortunately, P1 *rrnB* association with slow-growing mycobacteria is unusual, and more analyses are required to reveal its role. According to the consensus −10/−35 promoter elements [29], *M. kumamotonense* P1 *rrn*A promoter corresponds to a class B, while PCL1 promoter is grouped in class A, as well as the promoter P1 *rrn*B. As previously described, all of the mycobacterial sigma factors belonged to the σ^70^ family [30]. The −35 and −10 boxes of the *rrn*B single promoter were closely similar to the consensus sequence described for Sig A (Appendix A), the primary sigma factor of mycobacteria [31]. On the other hand, the association between sequences of the two *rrn*A promoters to sigma factor binding sites was less clear. This is in accordance with the lack of a defined consensus sequence for promoter boxes motifs of mycobacteria [31]. In fact, we could not find any similarity between the −35 and −10 boxes of the *rrn*A P1 promoter with any sigma factor sequence target. However, the −35 box sequence of *rrn*A PCL1 was found to be similar to the corresponding binding site of the *M. tuberculosis* SigD (Appendix A) [31]. Interestingly, by analyzing a SigD mutant of *M. tuberculosis*, Ares and co-workers (2017) found that the genes involved in septum formation contained the −35 and −10 boxes of that sigma factor. Indicating, that SigD could be also related to the synthesis of rRNA, which is in agreement with our result [32].

In addition, we have shown the organization of the promoter region of *rrnB* operon of *M. kumamotonense* (Figure 2A), which is absent in *M. tuberculosis* and only has 18% coverage regarding *rrnA.* Hence, additional specific studies on *rrnB* are required to find the presence of specific sequences that could be used to make a correct differentiation between both species of mycobacteria.

### 4.2. Contribution of the rrn Promoters to rRNA Synthesis

Analyzing the relative abundance of transcripts generated from each promoter, it has been described that P1 *rrnA* promoters appear to be weaker than other promoters such as PCL1 [10]. Here, we have shown the contributions of *rrnA* and *rrnB* promoters to pre-rRNA synthesis under different stress conditions in *M. kumamotonense*. The results suggest that, under optimal conditions for balanced growth (exponential phase), the *rrnA* promoters are the main contributor to the rRNA synthesis, but the tandem promoters P1 *rrnA* and PCL1 are not equally used. At the exponential phase, the PCL1 promoter was the main contributor, these results make sense with the previous data described in other mycobacterial species [10]. The lower expression from P1 *rrnA* could be partially occluded in the tandem context of the organization (P1-PCL1), as previously associated in *E. coli* [33]. On the other hand, the contribution of the *rrnB* promoter to rRNA synthesis was lower than the one observed from *rrnA* promoters. A similar rate of contribution of each promoter was observed at the stationary phase showing a minimal dependency on the *rrnB* promoter activity.

The nutrient deprivation stage of a bacterial culture has been described as a stage with a lack of nutrients necessary for continuing growth [17]. We suggest that *M. kumamotonense* grown under a lack of nutrients use mainly the *rrnA* operon, and this preference could be related to its highly conserved CL1 motif, whereas the P1 *rrnB* promoter is less active; this promoter probably needs additional factors to initiate the transcription as well its participation could be dependent on specific growth circumstances, such as hypoxic conditions, where the stability of rRNA could be compromised [34].

This study showed that *M. kumamotonense* survives under the two stages of non-replicating persistence of the Wayne and Hayes model NRP1 and NRP2 [13]. During these stress conditions, this bacterium promotes the upregulation of genes such as alkyl hydroperoxide reductase C (*ahpC*), a gene induced by hypoxia [35]; the protein encoded by this gene could be participating in the maintenance of redox homeostasis to face the low oxygen conditions. While the upregulation of *groEL,* a member of the hypoxic signature gene of *M. tuberculosis* [36] that codifies for a chaperon protein during the NRP1 stage, supports the expression of an adaptive response to stress in *M. kumamotonense.* This indicates the capacity of these bacteria to establish a dormancy-like phenotype and putatively generate latent infections.

At the NRP1 hypoxia phase, the contribution of the *rrnB* promoter to pre-rRNA synthesis increases at least twice compared with the exponential phase. Interestingly, when the mycobacterium reached NRP2, the anaerobiosis phase, the contribution of P1 *rrnB* decreased (Figure 3). The overexpression of pre-*rrn* products from the P1 *rrnB* promoter at NRP1 might play an important role during the adaptation process to the lower oxygen conditions faced by *M. kumamotonense* at this stage. However, the low level of P1 *rrnB* associated with the stress-response to anaerobiosis in NRP2 suggests that, at this phase, *M. kumamotonense* needs low metabolic activity and a low replication rate [37,38]; consequently, the rRNA requested for its persistence could be lower.

The fact that mycobacteria are primarily acquired via the respiratory route makes the epithelial cell models of the respiratory tract of interest to describe the bacterial interaction with the host [19]. Our results suggest, during the interaction of *M. kumamotonense* with alveolar cells, a possible role of *rrnB* to maintain the rRNA expression required during those initial steps of stress conditions. Compared with exponential and nutrient deprivation conditions, we found overexpression of pre-*rrn* products from P1 *rrnB* after two hours of interaction with alveolar cells. Further analyses at different intracellular times are required to evaluate if expression levels change over longer periods of time.

We showed that *M. kumamotonense* is minimally dependent on the *rrnB* promoter at the stationary and NRP2 phases. There are insufficient data to explain why P1 *rrnB* contributes so weakly to pre-rRNA synthesis. Nevertheless, when *M. kumamotonense* entered stress conditions, the NRP1 (hypoxia) phase or interaction with alveolar cells (only for 2 h), we observed a greater contribution of P1 *rrnB* to pre-rRNA synthesis, indicating a possible role of *rrnB* during the adaptation process to face stress conditions necessary to promote its survival. These results provide new insights into pre-rRNA synthesis in mycobacteria, as well as the potential ability of *M. kumamotonense* to establish a latent infection.

## Figures and Tables

**Figure 1 genes-14-01023-f001:**
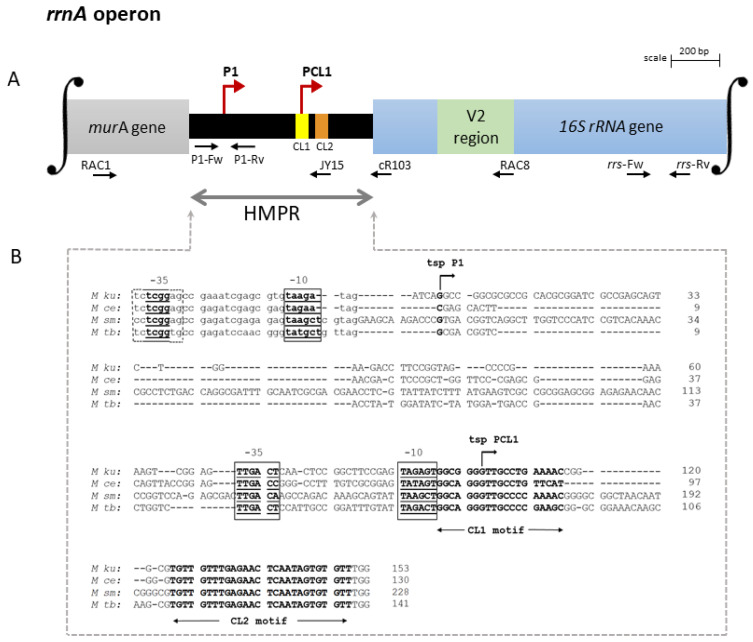
**Organization of the *rrnA* operon of *M. kumamotonense.*** (**A**) Schematic representation of the promoter region of the *rrnA* operon, which has two promoters, named P1 *rrnA* and PCL1, marked with red arrows. Promoter P1 *rrnA* was found downstream from the end of the *murA* coding region near the stop codon, whereas PCL1 was located within the HMPR. Horizontal arrows show the binding sites of the RAC1, RAC8, cR103, JY15, P1, and *rrs* primers. (**B**) Comparison of the nucleotide sequences of the HMPR regions of *M. kumamotonense* (*M ku*), *M. celatum* (*M ce*), *M. smegmatis* (*M sm*), and *M. tuberculosis* (*M tb*). The partial 3′-end of the *murA* gene is shown in lowercase letters. Putative-10 and -35 regions are shown within boxes in bold and underlined. The transcription starting point (*tsp*) of P1 *rrnA* is marked with an arrow and in bold located downstream from the end of *mur*A near the stop codon. The tsp of PCL1 is marked with an arrow and in bold located near to the conserved leader sequence-1 (CL1 motif) shown in bold, in the same way as the conserved leader sequence-2 (CL2 motif). −, deletion. Both tsp sites were established by 5′-RACE experiments. The distance between tspP1 and tspPCL1 was found to be 103 bp. The HMPR organization in *M. kumamotonense* and *M. tuberculosis* showed 93% similarity, compared to 84% and 97% with *M. celatum* and *M. smegmatis,* respectively. See the Section 2 for details and primer sequences. GenBank accession numbers: KT878832.1, EF613279.1, X87943.1, and X58890.1 for *M ku*, *M ce*, *M sm,* and *M tb*, respectively.

**Figure 2 genes-14-01023-f002:**
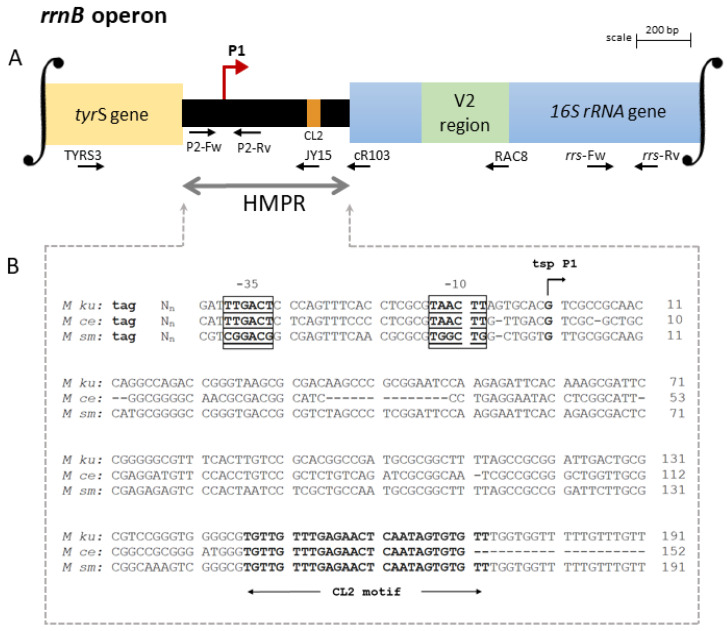
**Organization of the *rrnB* operon of *M. kumamotonense.*** (**A**) Schematic representation of the promoter region of the *rrnB* operon, which has a single promoter, named P1 *rrnB*, marked with a red arrow. The promoter P1 *rrnB* was located within the HMPR downstream from the end of the *tyr*S gene. Horizontal arrows show the binding sites of the TYRS3, RAC8, cR103, JY15, P2, and *rrs* primers. (**B**) Comparison of the nucleotide sequences of the HMPR regions of *M. kumamotonense* (*M ku*), *M. celatum* (*M ce*), and *M. smegmatis* (*M sm*). The stop codon for the *tyrS* gene is shown in lowercase letters. N_n_, specific sequence of each mycobacterium. Putative-10 and -35 regions are shown within boxes in bold and underlined. The transcription starting point (*tsp*) of P1 *rrnB* is marked with an arrow and in bold. The conserved leader sequence-2 (CL2 motif) is shown in bold. −, deletion. The tsp was established by 5′-RACE experiments. The HMPR organization in *M. kumamotonense* and *M. celatum* showed 78% similarity, while it was 83% similar to *M. smegmatis*. See the Section 2 for details and primer sequences. GenBank accession numbers: OQ344267, EF613280.1, and U09862.1 for *M ku*, *M ce*, and *M sm*, respectively.

**Figure 3 genes-14-01023-f003:**
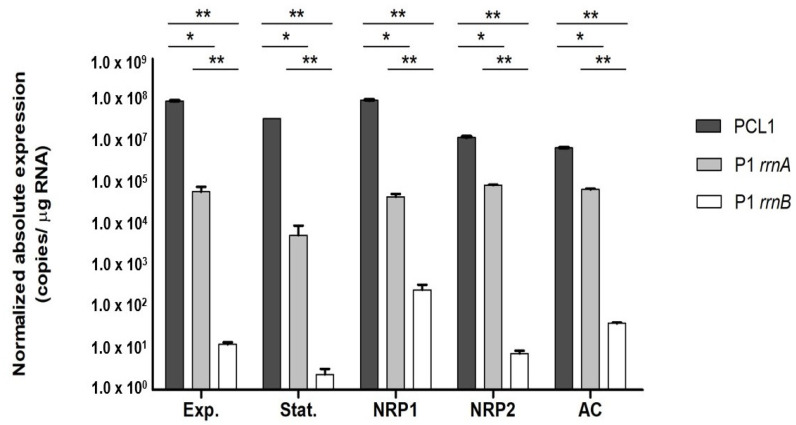
**Contribution of *rrnA* and *rrnB* promoters to rRNA synthesis in *M. kumamotonense* grown under different stress conditions.** Each bar represents the absolute quantification of the amounts of pre-*rrn* products from each promoter PCL1, P1 *rrnA*, or P1 *rrnB*, as relates to their contribution to the expression of 16S RNA, under exponential, stationary, NRP1, NRP2, and alveolar cell infection. See Appendix A for primers used. To obtain the specific number of copies of PCL1, the expression obtained from P1 *rrnA* and P1 *rrnB* was subtracted to the expression of 16S RNA (see the Section 2 for details). Exp., exponential; Stat., stationary; NRP1, non-replicative persistence 1; NRP2, non-replicative persistence 2; AC, alveolar cell infection. * *p* < 0.05; ** *p* < 0.005.

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
