# Peer review of "Organization and Characterization of the Promoter Elements of the rRNA Operons in the Slow-Growing Pathogen Mycobacterium kumamotonense"

_genes, 2023, doi:10.3390/genes14051023_

Round 1

Reviewer 1 Report

I would like to congrats the authors for their high-quality research on the expression and functional characterization of promoter elements in the rrnA and rrnB operons related to the growth rate and infection of the M. kumamotonense strain. 

I want to point out some suggestions.

1. Add the agarose gel with the amplicon products as corroboration and quality for the PCRs in the supplementary figures.

2. In 2.3 of M&M, the authors should mention if the primers were designed with the restriction enzyme sequences within the primer sequences or if the products were treated with restriction enzymes after for palindromes regions. If so, I would suggest bolding the enzyme regions in the sequences of the primers (Table S1) and mentioning which of these were used for specific ligation to pTZ57R/T plasmid, as isn't mentioned, for a more detailed description.

 3. please clarify the initial bacterial concentration used for the infection, in 2.4  as mentioned, an MOI of 100:1 according to the initial amount of cell/well (1x10^6); the infection was performed with 1x10^9 CFU/ml???

4. Please add Figure S1 (growth curve) is not present in the supplementary figures as is mentioned in the manuscript.

5.

Author Response

Response to Reviewer 1 Comments:

Point 1. Add the agarose gel with the amplicon products as corroboration and quality for the PCRs in the supplementary figures.

Reply: Images of agarose gels with the amplicons generated form each TSPs in RACE experiments were included in the new Supplementary Figure S2. And we referenced it, line 178.

Point 2. In 2.3 of M&M, the authors should mention if the primers were designed with the restriction enzyme sequences within the primer sequences or if the products were treated with restriction enzymes after for palindromes regions. If so, I would suggest bolding the enzyme regions in the sequences of the primers (Table S1) and mentioning which of these were used for specific ligation to pTZ57R/T plasmid, as isn't mentioned, for a more detailed description.

Reply: To carry out the cloning of the PCR products, the plasmid pTZ57R/T was used, which we can clone PCR products, through the T/A system, therefore, it was not necessary to design primers with restriction sequences within the primer neither the PCR products were treated with restriction enzymes. This information is now mentioned in line 120.

Point 3. please clarify the initial bacterial concentration used for the infection, in 2.4 as mentioned, an MOI of 100:1 according to the initial amount of cell/well (1x10^6); the infection was performed with 1x10^9 CFU/ml???

Reply: This information has been clarified in line 134; monolayers were infected with 1x108 CFU/well maintaining a MOI of 100:1 line 135.

Point 4. Please add Figure S1 (growth curve) is not present in the supplementary figures as is mentioned in the manuscript.

Reply: We apologize for this inconvenient, now this figure is included.

Reviewer 2 Report

The manuscript by Ricardo Sánchez-Estrada et al. described the Organization and characterization of the promoter elements of 2 the rRNA operons in the slow-growing pathogen Mycobacterium kumamotonense.

The manuscript is a brief report describing the organization of the operon with some characterization. I have a few concerns, which authors can consider for improving the ms. 

(i) Figure 1 needs the RACE data. Gel pictures showing the PCR products are required.

(ii) Is the position of TSPs of rrnA and rrnB identified by RACE is conserved across different mycobacterial species?

(iii) More information is needed in the description of promoters. The -10 and -35 sequences within rrnA and rrnB of Mycobacterium kumamotonenseare genes are similar to consensus sequences for which sigma factors? 

(iv) In the qRT_PCR experiments, positive controls are needed to establish that the stress conditions are properly established. For example, dosR-dependent genes for NRP1/NRP2. 

(v) Discussion is overly descriptive. The discussion should be crisp and to the point for a brief report.

Author Response

Response to Reviewer 2 Comments:

Point 1. Figure 1 needs the RACE data. Gel pictures showing the PCR products are required.

Reply: Images of agarose gels with the amplicons generated form each TSPs in RACE experiments were included in the new Supplementary Figure S2. And we referenced it, line 178.

Point 2. Is the position of TSPs of rrnA and rrnB identified by RACE is conserved across different mycobacterial species?

Reply: Yes, as previously described by Gonzalez-y-Merchand et al., 1997, the TSPs of rrnA are shown highly conserved between fast and slow growers’ mycobacterial species, mainly P1 and PCL1. And the localization of these TSPs appear to be related with the location of the murA gene and CL1 motif respectively, as it is shown in the figure 1 of this work, the TSP of P1 is located immediately downstream the ORF of murA whereas PCL1 appears upstream of PCL1 motif. In the other case, P1-rrnB has been described as the only one TSP of rrnB operon, and in the all-mycobacterial species in which organization has been described (Gonzalez-y-Merchand et al., 1996; Menendez et al., 2002) shows a conserved localization between the tyrS gene and CL2 motif as shown in figure 2 of the present work.

Point 3. More information is needed in the description of promoters. The -10 and -35 sequences within rrnA and rrnB of Mycobacterium kumamotonenseare genes are similar to consensus sequences for which sigma factors?

Reply: According to the sequences of the -35 and -10 boxes found in the M. kumamotonense rrn promoters, a searching for putative associated sigma factors were undertaken. And we described this interaction in lines 367-382.

According to the consensus −10/−35 promoter elements [29], M. kumamotonense P1 rrnA promoter corresponds to a class B while PCL1 promoter is grouped in class A, as well as the promoter P1 rrnB. As described previously, all the mycobacterial sigma factors belonged to the σ70 family [30]. The -35 and -10 boxes of the rrnB single promoter, was closely similar to the consensus sequence described for Sig A (Table S2), the primary sigma factor of mycobacteria [31]. On the other hand, the association between sequences of the two rrnA promoters to sigma factors binding sites was less clear. This is in accordance with the lack of a defined consensus sequence for promoter boxes motifs of mycobacteria [31]. In fact, we could not find similarity between the -35 and -10 boxes of the rrnA P1 promoter with any sigma factor sequence target. However, the -35 box sequence of rrnA PCL1 was found to be similar to the corresponding binding site of the M. tuberculosis SigD (Table S2)[31]. Interestingly, by analyzing a SigD mutant of M. tuberculosis, Ares and co-workers (2017) found that the genes involved in septum formation contained the -35 and -10 boxes of that sigma factor. Indicating, that SigD could be also related with the synthesis of rRNA, which is in agreement with our result [32]

Point 4. In the qRT_PCR experiments, positive controls are needed to establish that the stress conditions are properly established. For example, dosR-dependent genes for NRP1/NRP2.

Reply: We determined the expression of genes induced by hypoxia Rv3417c (groEL) and Rv2428 (ahpC) in order to corroborate the properly established NRP1 and NRP2 conditions, as kindly suggested by the reviewer, this information is shown in lines 90-93; 295-299 and 429-435, as well as in the supplementary figure 3.

Point 5. Discussion is overly descriptive. The discussion should be crisp and to the point for a brief report.

Reply: Discussion has been reduced in order to focus on the main points as reviewer recommended. Additionally, a substantial English spelling check of the manuscript was done.

Point 6. Quality of English Language

Reply: The language and grammar of the manuscript have been professionally reviewed, the review was carried out by Proof-Reading-Service, the certificate of this Enterprise is attached.

Reviewer 3 Report

Organization and characterization of the promoter elements of 2 the rRNA operons in the slow-growing pathogen Mycobacterium kumamotonense 4

Ricardo Sánchez-Estrada et al.

In this brief report, the authors show the organization of the promoter elements in the rrnA and rrnB operons of M. kumamotonensis, the DNA sequences of the intergenic regions present in both operons, and evaluate how different stress conditions (lack of nutrients, hypoxia, and cellular infection) affect the expression of 16S rRNA.

The experiments are well performed and the results are interpreted appropriately.

The relevance of this work is that the promoter region of rrnB operon of M. kumamotonense, which is absent in M. tuberculosis and only has 54% of coverage regarding to rrnA, could be used to make a correct differentiation between both species of mycobacteria.

Minor comments

Materials and methods

Lines 185-189 It will be useful to show the primers used in the figure (either in figures 1 and 2 or in a new supplementary figure along with the Table S1)

Results

Figure 1 (legend) line 262 CL1 motif it is not in italics.

“in the same way as the conserved leader sequence -2 (CL2 motif)”, it is not in italics either

Lines 293-294 “Compared with exponential conditions, the pre-rrn products coming from P1 rrnB promoter increased slightly, while PCL1 pre-rrn products decreased (Figure 3)”, Are these differences statically significant?

Discussion

Paragraph from line 373-380 is too long.

We suggest that M. kumamotonense grown under a lack of nutrients using mainly the rrnA operon, could be explained because this operon is highly conserved in mycobacteria mainly in the CL1 motif, which could be enhancing the transcription rates from this promoter, whereas the P1 rrnB promoter in the rrnB operon is less active. More studies are necessary to evaluate if this promoter may need additional factors to initiate the transcription of pre-rrn products as well as if its participation is dependent on specific growth circumstances such as hypoxic conditions, where the stability of rRNA could be compromised

Author Response

Response to Reviewer 3 Comments:

Point 1. Materials and methods

  • Lines 185-189 It will be useful to show the primers used in the figure (either in figures 1 and 2 or in a new supplementary figure along with the Table S1)

Reply: All primers used are show now in figures 1 and 2 as kindly suggested by the reviewer.

Point 2. Results

  • Figure 1 (legend) line 262 CL1 motif it is not in italics.

“in the same way as the conserved leader sequence -2 (CL2 motif)”, it is not in italics either

Reply: We have now amended it on, we eliminated “and italics” and only conserved “is shown in bold” as description of the CL1 and CL2 motifs lines 262 and 278.

  • Lines 293-294 “Compared with exponential conditions, the pre-rrn products coming from P1 rrnB promoter increased slightly, while PCL1 pre-rrn products decreased (Figure 3)”, Are these differences statically significant?

Reply: Yes, now this information is mentioned in the line 315

Point 3. Discussion

  • Paragraph from line 373-380 is too long.

We suggest that M. kumamotonense grown under a lack of nutrients using mainly the rrnA operon, could be explained because this operon is highly conserved in mycobacteria mainly in the CL1 motif, which could be enhancing the transcription rates from this promoter, whereas the P1 rrnB promoter in the rrnB operon is less active. More studies are necessary to evaluate if this promoter may need additional factors to initiate the transcription of pre-rrn products as well as if its participation is dependent on specific growth circumstances such as hypoxic conditions, where the stability of rRNA could be compromised.

Reply: This paragraph was reduced as kindly suggested by the reviewer, now it reads

We suggest that M. kumamotonense grown under a lack of nutrients using mainly the rrnA operon, this preference could be related to its highly conserved CL1 motif, whereas the P1 rrnB promoter is less active, probably this promoter needs additional factors to initiate the transcription as well its participation could be dependent on specific growth circumstances such as hypoxic conditions, where the stability of rRNA could be compromised [31], lines 405-427.